# WEAKLY SUPERVISED MONOCULAR 3D DETECTION WITH A SINGLE-VIEW IMAGE

## ABSTRACT

Monocular 3D detection (M3D) aims for precise 3D object localization from a single-view image which usually involves labor-intensive annotation of 3D detection boxes. Weakly supervised M3D has recently been studied to obviate the 3D annotation process by leveraging many existing 2D annotations, but it often requires extra training data such as LiDAR point clouds or multi-view images which greatly degrades its applicability and usability in various applications. We propose SKD-WM3D, a weakly supervised monocular 3D detection framework that exploits depth information to achieve M3D with a single-view image exclusively without any 3D annotations or other training data. One key design in SKD-WM3D is a self-knowledge distillation framework, which transforms image features into 3D-like representations by fusing depth information and effectively mitigates the inherent depth ambiguity in monocular scenarios with little computational overhead in inference. In addition, we design an uncertainty-aware distillation loss and a gradient-targeted transfer modulation strategy which facilitate knowledge acquisition and knowledge transfer, respectively. Extensive experiments show that SKD-WM3D surpasses the state-of-the-art clearly and its performance is even on a par with many fully supervised methods.

## 1 INTRODUCTION

Monocular 3D detection (M3D) has emerged as one key component in the area of autonomous driving and computer vision. Its primary target is to recognize objects and obtain their 3D localization from single-view images. Thanks to its low deployment cost, M3D (Chen et al., 2016; Peng et al., 2022b) has attracted increasing attention in both academic and industrial sectors, achieving very impressive progress in recent years. On the other hand, most existing studies (Ku et al., 2019; Simonelli et al., 2020; Reading et al., 2021; Peng et al., 2022a) adopt a fully supervised setup which have been facing increasing scalability concern as large-scale 3D boxes are often labor-intensive to collect. Effective M3D training without 3D annotations has become a critical issue while handling M3D problems in various research and practical tasks.

Weakly supervised M3D (WM3D) (Peng et al., 2022b) has recently been explored for learning effective 3D detectors without 3D box annotations, aiming to exploit 2D annotations to make up for the absence of 3D information. For example, WeakM3D (Peng et al., 2022b) exploits LiDAR point clouds to infer 3D information as illustrated in Fig.1 (a). However, it requires costly and complicated LiDAR sensors to collect point clouds which limits its applicability and usability greatly. WeakMono3D (Tao et al., 2023) employs 2D information only by either leveraging multi-view stereo with images from multiple cameras or constructing pseudo-multi-view perspective from sequential video frames as illustrated in Fig.1 (b). However, collecting multi-view images is complicated, and resorting to a pseudo multi-view perspective degrades the detection performance clearly. With the advance of single-view depth estimation, WM3D with depth from a single-view image presents a potential solution for compensating the absence of 3D annotations. On the other hand, direct integration of such depth into existing frameworks often necessitates complex network architectures which further incurs significant computational costs. This gives rise to a pertinent question: When not using additional LiDAR point clouds or multi-view image pairs, is it possible to harness the depth from off-the-shelf depth estimators without introducing much computational overhead in inference?

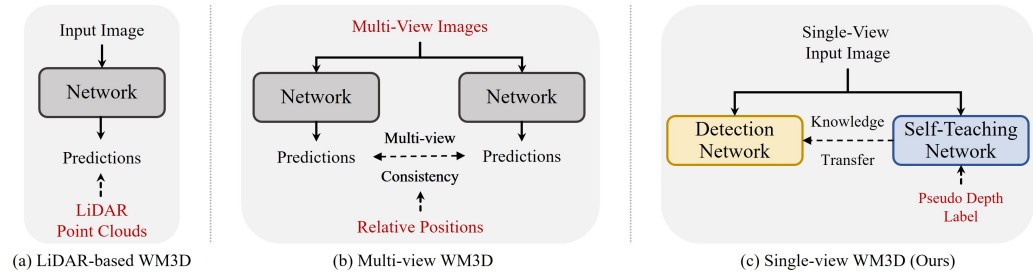

Figure 1: Comparison of high-level paradigms in weakly supervised monocular 3D detection. Our approach in (c) leverages *Pseudo Depth Labels* from a single-view image to achieve weakly supervised monocular 3D detection, requiring no extra training data like LiDAR point clouds or multiview images as in (a) and (b) and improving the usability and applicability greatly. Data in red denotes extra data in network training.

We design SKD-WM3D, a novel weakly supervised monocular 3D object detection method that is exclusively grounded on single-view images. One key design in SKD-WM3D is a self-knowledge distillation framework which consists of a **D**epth-guided **S**elf-teaching **N**etwork (DSN) and a **M**onocular 3D **D**etection **N**etwork (MDN). As illustrated in Fig. 1 (c), SKD-WM3D utilizes depth information to enhance the 3D localization ability of DSN and transfers such ability to MDN via self-knowledge distillation. Such self-distillation design enables MDN to unearth the intrinsic depth information from single-view images independently, bypassing additional modules such as pre-trained depth estimation networks and leading to precise and efficient 3D localization with little computational overhead during inference. On top of DSN and MDN, we design an uncertainty-aware distillation loss to optimize the utilization of the transferred knowledge by weighting up more certain knowledge while weighting down less certain knowledge. In addition, we design a gradient-targeted transfer modulation strategy to synchronize the learning paces of DSN and MDN by prioritizing MDN learning at the initial stage when MDN lags behind DSN and enabling it to provide more feedback to DSN when MDN is better trained at late stages.

Our contribution can be summarized in three aspects. *First*, we design a novel framework that achieves weakly supervised monocular 3D detection by distilling knowledge between a depth-guided self-teaching network and a monocular 3D detection network. Without any extra training data like LiDAR point clouds or multi-view images, the framework exploits depth exclusively from a single image with little computational overhead in inference. *Second*, we design an uncertainty-aware distillation loss and a gradient-targeted transfer modulation strategy which facilitate knowledge acquisition and knowledge transfer, respectively. *Third*, the proposed approach clearly outperforms the state-of-the-art in weakly supervised monocular 3D detection, and its performance is even on par with several fully supervised methods.

## 2 RELATED WORK

### 2.1 MONOCULAR 3D DETECTION

Monocular 3D object detection aims to predict 3D object localization from single-view images. Standard monocular detectors (He & Soatto, 2019; Brazil & Liu, 2019; Chen et al., 2020; Zhou et al., 2021; Zhang et al., 2023) operate solely on single images, without utilizing additional data. However, the inherent depth ambiguity of monocular detection significantly hinders its performance compared to its stereo counterparts. To address this limitation, various approaches seek solutions with the help of extra data, such as LiDAR point clouds (Ku et al., 2019; Ma et al., 2019; Chen et al., 2021; Chong et al., 2022), video sequences (Brazil et al., 2020), 3D CAD models (Chen et al., 2016; Liu et al., 2021; Murthy et al., 2017), and depth estimation (Ding et al., 2020; Qin et al., 2019; Wang et al., 2019; You et al., 2020). Specifically, MonoRUn (Chen et al., 2021) adopts an uncertainty-aware regional reconstruction network for regressing pixel-associated 3D object coordinates with LiDAR point clouds as extra supervision. MonoDistill (Chong et al., 2022) introduces an effective distillation-based approach that incorporates spatial information from LiDAR signals into monocular

3D detection. Additionally, pseudo-LiDAR-based methods (Wang et al., 2019; You et al., 2020) convert estimated depth maps to simulate the real LiDAR point clouds to utilize the well-designed LiDAR-based 3D detector. During inference, compared with methods using depth estimation, our method eliminates the need for pseudo depth labels and complex network architectures, with little computational overhead. Besides, existing fully supervised methods require large-scale 3D box ground truth, which is labor-intensive to collect and annotate.

## 2.2 WEAKLY SUPERVISED 3D OBJECT DETECTION

Due to the high cost of annotating 3D boxes in the 3D object detection task, various weakly supervised approaches have been proposed. For example, WS3D (Meng et al., 2020) presents a weakly supervised method for 3D LiDAR object detection, which requires only a limited number of weakly annotated scenes with center-annotated BEV maps. VS3D (Qin et al., 2020) introduces a cross-model knowledge distillation strategy to transfer the knowledge from the RGB domain to the point cloud domain, using LiDAR point clouds as weak supervision. Recent research on weakly supervised 3D object detection has turned to exploring the monocular setting. For example, WeakM3D (Peng et al., 2022b) generates 2D boxes to select RoI LiDAR point clouds as weak supervision and then predicts 3D boxes that closely align with the selected RoI LiDAR point clouds. More recently, WeakMono3D (Tao et al., 2023) eliminates the need for LiDAR, offering both multi-view and single-view yet multi-frame versions. While the former acquires stereo image inputs from multiple cameras, the latter constructs a pseudo-multi-view perspective using video frames, suffering from clear performance degradation. Instead of requiring extra training data like LiDAR point clouds or multi-view images, we tackle the challenge of weakly supervised monocular 3D detection by leveraging a single-view image exclusively.

## 2.3 SELF-KNOWLEDGE DISTILLATION

Knowledge distillation (Hinton et al., 2015; Liu et al., 2019b; Park et al., 2019; Tian et al., 2019; Romero et al., 2014; Chung et al., 2020; Zhao et al., 2022; Huang et al., 2022) transfers knowledge from a pre-trained teacher network to a student network for improving its performance. Self-knowledge distillation (Szegedy et al., 2016; Müller et al., 2019; Yang et al., 2023), distinct from traditional knowledge distillation, leverages the information within the student model to facilitate its learning without the pre-trained teacher network. Specifically, data augmentation approach (Xu & Liu, 2019; Yun et al., 2020; Heo et al., 2019) transfers knowledge through different distortions of the same training data. However, they are susceptible to inappropriate augmentations, such as improper instance rotation or distortion, potentially introducing noise that hampers network learning. Another typical approach exploits auxiliary networks (Zhu et al., 2018; Zhang et al., 2019). For example, DKS (Sun et al., 2019) introduces auxiliary supervision branches and pairwise knowledge alignments, while FRSKD (Ji et al., 2021) adds a new branch supervised by the original features and utilizes both soft-label and feature-map distillation. Our work is the first that introduces self-knowledge distillation with auxiliary networks for weakly supervised monocular 3D detection. It effectively exploits depth information from a single-view image with little computational overhead during inference.

## 3 METHODOLOGY

This section presents the proposed SKD-WM3D. First, the problem definition and overview are presented in Sec. 3.1. Then detailed designs of SKD-WM3D are introduced, including the self-knowledge distillation framework in Sec. 3.2, the uncertainty-aware distillation loss in Sec. 3.3 and the gradient-targeted transfer modulation in Sec. 3.4. Finally, loss functions are presented in Sec. 3.5.

## 3.1 PROBLEM DEFINITION AND OVERVIEW

Weakly supervised monocular 3D detection takes an RGB image as input, intending to classify objects and determine the corresponding bounding boxes in 3D space without using 3D box annotations. The prediction of each object is composed of the object category $C$, a 2D bounding box $B^{2D}$, and a 3D bounding box $B^{3D}$. Specifically, the 3D box $B^{3D}$ can be further decomposed to the object

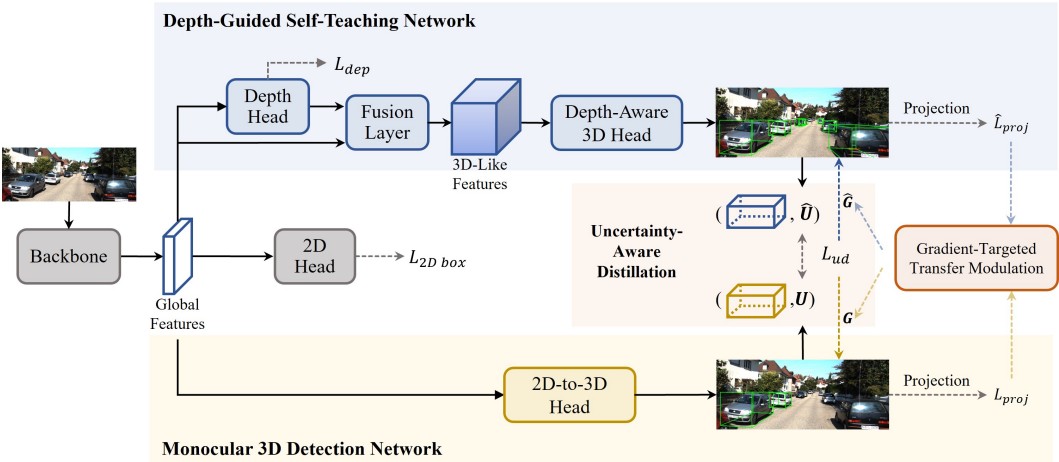

Figure 2: The framework of the proposed self-knowledge distillation network. The framework consists of a depth-guided self-teaching network and a monocular 3D detection network. The depth-guided self-teaching network acquires comprehensive 3D localization knowledge by leveraging depth information and transfers its learned expertise to the monocular 3D detection network via soft label distillation to enhance its performance. We design an uncertainty-aware distillation loss and a gradient-targeted transfer modulation strategy to facilitate the knowledge transfer between the two networks effectively. During inference, the monocular 3D detection network extracts intrinsic depth information from single-view images independently with little computational overhead.

3D location $(x_{3D}, y_{3D}, z_{3D})$, the object dimension with height, width and length $(h_{3D}, w_{3D}, l_{3D})$, as well as orientation $\theta$.

We design a self-knowledge distillation framework to tackle the challenge of weakly supervised monocular 3D detection from a single-view image. As Fig. 2 shows, the framework consists of two subnetworks including a *Depth-Guided Self-Teaching Network* and a *Monocular 3D Detection Network*. In the *Depth-Guided Self-Teaching Network*, the global features $F_G$ extracted by the backbone are fed into a *Depth Head* to obtain depth features. Next, the global features $F_G$ and the extracted depth features are fed into a *Fusion Layer* to obtain 3D-like features $F_{3D}$. Then 3D box $\widehat{B}_p^{3D}$ and uncertainty $\widehat{U}$ are predicted by a *Depth-Aware 3D Head* with 3D-like features as the input. In the *Monocular 3D Detection Network*, the global features $F_G$ are first fed into a *2D-to-3D Head* to predict 3D box $\widehat{B}_p^{2D}$ and uncertainty $U$. Besides, the 3D boxes predicted by both networks are further projected into 2D boxes. Moreover, we design an uncertainty-aware distillation loss $L_{ud}$ to obtain low-uncertainty knowledge, and a gradient-targeted transfer modulation strategy to synchronize the learning paces between the two networks by controlling gradients $\widehat{G}$ and $G$ of $L_{ud}$.

## 3.2 SELF-KNOWLEDGE DISTILLATION FRAMEWORK

The self-knowledge distillation framework enhances the 3D localization ability of the depth-guided self-teaching network by utilizing depth information from an off-the-shelf depth estimator and then transfers the ability to the monocular 3D detection network via self-knowledge distillation.

**Depth-Guided Self-Teaching Network.** To equip the self-teaching network with 3D localization ability, we propose to learn from global features $F_G$ and depth information from an off-the-shelf depth estimator to acquire comprehensive 3D knowledge. The depth information is exploited via two major designs. Firstly, we introduce a depth head $\mathcal{D}$ that extracts depth features $F_D$ as follows:

$$F_D = \mathcal{D}(F_G), \tag{1}$$

The depth features $F_D$ are exploited to generate depth maps $D_p$, where the depth map generation is supervised by the pseudo ground truth of the depth map $D_{gt}$ that is predicted by an off-the-shelf

depth estimator by using the focal loss (Lin et al., 2017) as depth loss $L_{dep}$. Hence, the depth features can be acquired by the depth-guided self-teaching network effectively.

Secondly, we obtain 3D-like features $F_{G3D}$ by integrating the depth features $F_D$ that provide information along the depth dimension, as well as the global features $F_G$ that capture knowledge about the 2D image plane. Specifically, we design a fusion layer that fuses the depth features $F_D$ with the global features $F_G$ to derive the $F_{G3D}$ as follows:

$$F_{G3D} = FFN(CrossAttention(SelfAttention(F_D), F_G)), \quad (2)$$

where the $FFN$ is the feed-forward network, and the structures of $CrossAttention$ and $SelfAttention$ employ the standard transformer architecture (Vaswani et al., 2017). The obtained 3D comprehension improves the network's ability to precisely locate objects, effectively mitigating depth ambiguity arising from single-view image input.

**Monocular 3D Detection Network.**  The monocular 3D detection network acquires the 3D localization knowledge from the depth-guided self-teaching network. By distilling soft labels generated by the depth-guided self-teaching network, the monocular 3D detection network can extract intrinsic depth information from images independently during inference. This kills the need for additional complex modules such as pre-trained depth estimation networks or depth fusion modules, facilitating the inference with little computational overhead.

### 3.3 Uncertainty-Aware Distillation Loss

During the knowledge distillation process, uncertain knowledge could affect the network training negatively if all transferring knowledge is treated equally. To benefit more from certain knowledge and weaken the effect of uncertain knowledge, we design an uncertainty-aware distillation loss between the 3D boxes that are predicted by the two networks in the self-knowledge distillation framework. The uncertainty-aware distillation loss exploits the prediction uncertainty to modulate the distillation loss magnitude as follows:

$$L_{ud} = \frac{L_d}{min((\widehat{U} + U)/2, \alpha)} + \left\| min(\frac{\widehat{U} + U}{2}, \alpha) \right\|^2, \quad (3)$$

where $\widehat{U}$ and $U$ are the uncertainties corresponding to the 3D box predicted by the two networks, $\left\| min(\frac{\widehat{U} + U}{2}, \alpha) \right\|^2$ is the L2 regularization, and $\alpha$ is a fixed value set to 0.1. $L_d$ denotes the basic distillations loss, and we employ the commonly used SmoothL1 (Girshick, 2015) loss to enforce the consistency between the 3D boxes predicted by the two networks. The SmoothL1 loss leaves a soft margin when computing the difference between the two 3D boxes:

$$L_d = \begin{cases} 0.5 \times (\widehat{B}_p^{3D} - B_p^{3D})^2/\gamma, & \text{if } |\widehat{B}_p^{3D} - B_p^{3D}| < \gamma \\ |\widehat{B}_p^{3D} - B_p^{3D}| - 0.5 \times \gamma, & \text{otherwise} \end{cases}, \quad (4)$$

where $\widehat{B}_p^{3D}$ and $B_p^{3D}$ are the predicted 3D boxes from the depth-guided self-teaching network and the monocular 3D detection network, respectively. $\gamma$ is the soft margin, which is set to 1.0.

**Remark.**  The proposed uncertainty-aware distillation loss $L_{ud}$ integrates average uncertainty $\frac{\widehat{U} + U}{2}$ as regularization and a weighted component for the basic distillation loss $L_d$, allowing adaptive learning adjustments based on the knowledge's uncertainty level. Specifically, when dealing with uncertain knowledge, a smaller weight is assigned to the basic distillation loss $L_d$ to mitigate potential adverse effects on network learning. Consequently, the network prioritizes optimizing uncertainty reduction in such scenarios. When dealing with certain knowledge, the network emphasizes optimizing the basic distillation loss $L_d$ due to its higher weight. Notably, the basic distillation loss $L_d$ simply considers box consistency, while integrating uncertainty is beneficial for enhancing the knowledge distillation process.

### 3.4 GRADIENT-TARGETED TRANSFER MODULATION STRATEGY

The depth-guided self-teaching network, which leverages depth information to predict 3D boxes, transfers its learned 3D knowledge to the monocular 3D detection network. The asynchronous learning paces of the two networks pose potential challenges to effective 3D knowledge transfer.

We design a gradient-targeted transfer modulation strategy to synchronize the learning pace of the depth-guided self-teaching network and the monocular 3D detection network. We modulate the knowledge transfer dynamically, by controlling the gradients from the uncertainty-aware distillation loss $L_{ud}$. Specifically, we adapt the gradients based on the 2D projection performance of each network, assigning smaller backward gradients for good-performing network and higher backward gradients for bad-performing network. The gradient-targeted transfer modulation strategy is formulated as follows:

$$\widehat{G}' = \frac{2 \times \widehat{L}_{proj}}{\widehat{L}_{proj} + L_{proj}} \times \widehat{G}, G' = \frac{2 \times L_{proj}}{\widehat{L}_{proj} + L_{proj}} \times G, \tag{5}$$

Where $\widehat{G}$ and $G$ are the original gradients of the two networks, $\widehat{G}'$ and $G'$ are the modified gradients, $\widehat{L}_{proj}$ and $L_{proj}$ are projection losses, computed between the projected 2D boxes from 3D box predictions and 2D box annotations.

The gradient-targeted transfer modulation prioritizes training the monocular 3D detection network when its learning lags behind the depth-guided self-teaching network at the early training stage. As the monocular 3D detection network learns and improves gradually, it is enabled to provide more feedback progressively to the depth-guided self-teaching network.

### 3.5 LOSS FUNCTIONS

The overall objective consists of three losses including $L_{ud}$, $L_{dep}$ and $L_{base}$. $L_{ud}$ is the uncertainty-aware distillation loss as defined in Sec. 3.3. $L_{dep}$ is the depth loss for supervising the predicted depth map. $L_{base}$ includes losses for supervising 2D boxes prediction by 2D heads and the 3D box predictions, which has been adopted in prior CenterNet (Zhou et al., 2019) and WeakMono3D (Tao et al., 2023). We set the weight for each loss item to 1.0, and the overall loss function can be formulated as follows:

$$L = L_{ud} + L_{dep} + L_{base}, \tag{6}$$

## 4 EXPERIMENTS

### 4.1 DATASET

We conduct experiments over the 3D KITTI dataset (Geiger et al., 2012) that has been widely adopted for benchmarking of 3D object detection methods. The dataset consists of 7,481 images for training and 7,518 images for testing. The labels of the train set are publicly available and the labels of the test set are stored on a test server for evaluation. For ablation studies, we follow (Chen et al., 2016) which divides the 7,481 training samples into a new train set with 3,712 images and a validation set with 3,769 images.

### 4.2 EVALUATION PROTOCOLS

Following (Simonelli et al., 2020), we adopt the evaluation metric $AP|_{R_{40}}$ which is the average of the AP of 40 recall points. We report the average precision on bird's eye view and 3D object detection as $AP_{BEV}|_{R_{40}}$ and $AP_{3D}|_{R_{40}}$. In addition, as most weakly supervised 3D object detection methods apply IoU threshold of 0.7 for the test set and 0.5 for the validation set, we adopt the same thresholds for fair benchmarking.

### 4.3 IMPLEMENTATION DETAILS

We conduct experiments on 2 NVIDIA V100 GPUs with batch size of 16, and train the framework with 150 epochs. We use the Adam optimizer with the initial learning rate $1e^{-5}$, which is gradually

Table 1: Comparison on the performance of the Car category on KITTI *test* set. For all results, we use $\text{AP}|_{R_{40}}$ metrics with IoU threshold equals to 0.7. The best results are in **bold**.

| Method | Supervision | $\text{AP}_{BEV}/\text{AP}_{3D}(\text{IoU}=0.7)|_{R_{40}}$ | | |
| --- | --- | --- | --- | --- |
| | | Easy | Moderate | Hard |
| WeakM3D (Peng et al., 2022b) | | 11.82/5.03 | 5.66/2.26 | 4.08/1.63 |
| WeakMono3D (Tao et al., 2023) | Weak | 12.31/6.98 | 8.80/4.85 | 7.81/4.45 |
| SKD-WM3D (Ours) | | **15.71/8.95** | **10.15/5.54** | **8.08/4.53** |

Table 2: Comparison on the performance of the Car category on KITTI *val* set. For all results, we use $\text{AP}|_{R_{40}}$ metric with IoU threshold equals to 0.5. * denotes this performance is reproduced from the official code. The best results of weakly supervised approaches are in **bold**.

| Method | Supervision | $\text{AP}_{BEV}/\text{AP}_{3D}(\text{IoU}=0.5)|_{R_{40}}$ | | |
| --- | --- | --- | --- | --- |
| | | Easy | Moderate | Hard |
| CenterNet (Zhou et al., 2019) | | 34.36/20.00 | 27.91/17.50 | 24.65/15.57 |
| MonoGRNet (Qin et al., 2019) | | 52.13/47.59 | 35.99/32.28 | 28.72/25.50 |
| M3D-RPN (Brazil & Liu, 2019) | | 53.35/48.53 | 39.60/35.94 | 31.76/28.59 |
| MonoPair (Chen et al., 2020) | | 61.06/55.38 | 47.63/42.39 | 41.92/37.99 |
| MonoDLE (Ma et al., 2021) | Full | 60.73/55.41 | 46.87/43.42 | 41.89/37.81 |
| GUPNet (Lu et al., 2021) | | 61.78/57.62 | 47.06/42.33 | 40.88/37.59 |
| Kinematic (Brazil et al., 2020) | | 61.79/55.44 | 44.68/39.47 | 34.56/31.26 |
| MonoDistill (Chong et al., 2022) | | 71.45/65.69 | 53.11/49.35 | 46.94/43.49 |
| MonoDETR (Zhang et al., 2023)* | | 72.34/68.05 | 51.97/48.42 | 46.94/43.48 |
| VS3D (Qin et al., 2020) | | 31.59/22.62 | 20.59/14.43 | 16.28/10.91 |
| Autolabels (Zakharov et al., 2020) | | 50.51/38.31 | 30.97/19.90 | 23.72/14.83 |
| WeakM3D (Peng et al., 2022b) | Weak | **58.20**/50.16 | 38.02/29.94 | 30.17/23.11 |
| WeakMono3D (Tao et al., 2023) | | 54.32/49.37 | 42.83/39.01 | 40.07/36.34 |
| SKD-WM3D (Ours) | | 55.47/**50.21** | **44.35/41.57** | **41.86/36.92** |

increased to $1e^{-3}$ for the first 5 epochs and decayed with rate 0.1 at the 90 and 120 epochs. We employ DLA-34 (Yu et al., 2018) as the detector's backbone. The pseudo ground truth of the depth map is generated with an off-the-shelf depth estimator without using the ground truth of depth label.

## 4.4 COMPARISON WITH STATE-OF-THE-ART METHODS

We compare our method with several state-of-the-art weakly supervised monocular 3D detection methods on the KITTI test set. As Table 1 shows, our method achieves superior detection performance across all metrics. This superior performance is largely attributed to our designed self-knowledge distillation framework that extracts and exploits intrinsic depth information from a single-view image effectively. It should be highlighted that our method employs a single-view image exclusively without involving additional training data such as LiDAR point clouds (Peng et al., 2022b) or multi-view image pairs (Tao et al., 2023).

Table 2 shows the benchmarking on the KITTI validation set. Specifically, we compare our method against both state-of-the-art weakly supervised monocular 3D detection methods and fully supervised methods. It can be seen that our method achieves superior performance compared to WeakM3D (Peng et al., 2022b) and WeakMono3D (Tao et al., 2023) across most metrics. Additionally, its performance is even on a par with several fully supervised methods (Zhou et al., 2019; Qin et al., 2019; Brazil & Liu, 2019).

**Quantitative Results** Fig. 3 shows qualitative illustration with both 2D RGB images and 3D point clouds. In simple scenarios, our model achieves great prediction precision, which is largely attributed to the proposed self-knowledge distillation framework as well as the uncertainty-aware distillation loss and gradient-targeted transfer modulation strategy, all working together to facilitate comprehensive 3D information extraction effectively. But for heavily occluded or distant objects, the accuracy of orientation and depth estimation drops more or less which is common for monocular

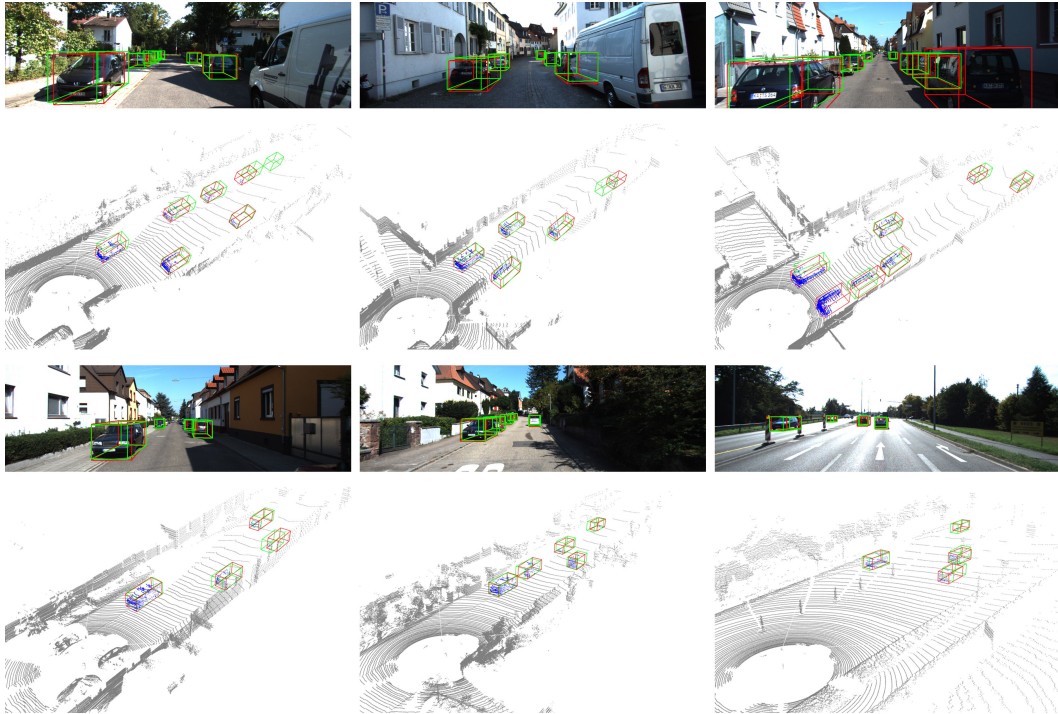

Figure 3: Qualitative illustration on KITTI *val* set. Red boxes denote ground-truth annotations and Green boxes denote our predictions. The ground truth of LiDAR point clouds is utilized for visualization purposes only. Best viewed with zoom-in.

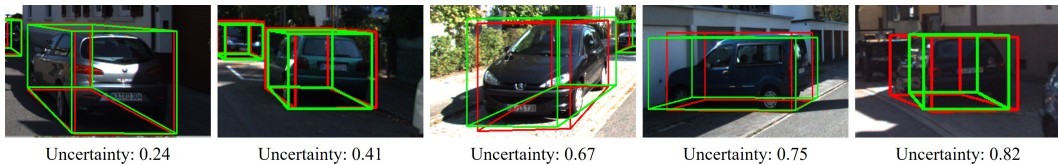

| Uncertainty: 0.24 | Uncertainty: 0.41 | Uncertainty: 0.67 | Uncertainty: 0.75 | Uncertainty: 0.82 |

Figure 4: Qualitative illustration of object detection and the corresponding detection uncertainties on KITTI *val* set. Red boxes denote ground-truth annotations and Green boxes denote our predictions. The detection accuracy is closely correlated with the detection uncertainty.

3D detection due to its ill-posed nature. In addition, we show the visualization of object detection and the detection uncertainties in Fig. 4. It can be observed that the prediction accuracy has a close correlation with the prediction uncertainty.

## 4.5 ABLATION STUDY

We conduct extensive ablation studies on the KITTI validation dataset to evaluate our designs. Specifically, we evaluated the efficacy of the two individual networks in the proposed self-knowledge distillation framework. In addition, we examine the effect of the proposed uncertainty-aware distillation loss and the gradient-targeted transfer modulation strategy. Further, we evaluated the efficiency of our monocular 3D detection framework.

**Self-Knowledge Distillation Framework.** We train two models to assess the contributions of the two networks in our proposed self-knowledge distillation framework. As Table 3 shows, training the monocular 3D detection network alone produces few meaningful detection results as the absence of depth information leads to ambiguous object localization along the depth dimension. As a compari-

Table 3: Ablation study of the proposed self-knowledge distillation framework. The best results are in **bold**.

| Index | Monocular 3D Detection Network | Depth-Guided Self-Teaching Network | $\text{AP}_{BEV}/\text{AP}_{3D}(\text{IoU}=0.5)|_{R_{40}}$ | | |
|---|---|---|---|---|---|
| | | | Easy | Moderate | Hard |
| 1 | ✓ | | 0.00/0.00 | 0.00/0.00 | 0.00/0.00 |
| 2 | | ✓ | 45.23/40.96 | 34.27/31.02 | 30.17/26.27 |
| 3 | ✓ | ✓ | **55.47/50.21** | **44.35/41.57** | **41.86/36.92** |

Table 4: Ablation study of the proposed uncertainty-aware distillation loss and the gradient-targeted transfer modulation strategy. The best results are in **bold**.

| Index | Uncertainty-Aware Distillation Loss | Transfer Modulation Strategy | $\text{AP}_{BEV}/\text{AP}_{3D}(\text{IoU}=0.5)|_{R_{40}}$ | | |
|---|---|---|---|---|---|
| | | | Easy | Moderate | Hard |
| 1 | | | 49.95/44.61 | 38.24/35.74 | 37.28/34.82 |
| 2 | ✓ | | 53.16/48.13 | 41.85/39.02 | 40.14/35.70 |
| 3 | | ✓ | 52.35/46.30 | 41.45/38.91 | 39.73/35.44 |
| 4 | ✓ | ✓ | **55.47/50.21** | **44.35/41.57** | **41.86/36.92** |

Table 5: Comparison on inference time of methods utilizing dense depth maps.

| Method | PatchNet (Ma et al., 2020) | D4LCN (Ding et al., 2020) | DDMP-3D (Wang et al., 2021) | MonoDistill (Chong et al., 2022) | SKD-WM3D (Ours) |
|---|---|---|---|---|---|
| Runtime | 400ms | 200ms | 180ms | 40ms | 33ms |

son, training the depth-guided self-teaching network alone can produce reasonable detection results thanks to the estimated depth map pseudo labels. In addition, training both subnetworks concurrently produces the best 3D detection, validating the effectiveness of extracting 3D information from a single image. We can also see that including the self-knowledge distillation on top of the depth-guided self-teaching network greatly improves the detection by reducing the adverse effects of uncertain knowledge and enabling the communication between the two subnetworks during training.

**Uncertainty-Aware Distillation Loss and Gradient-Targeted Transfer Modulation.**  Table 4 shows the ablation study of the uncertainty-aware distillation loss and the gradient-targeted transfer modulation. We can observe that the baseline does not perform well due to the adverse effect of uncertain knowledge and the asynchronous learning paces of the two subnetworks. On top of the baseline, including either the uncertainty-aware distillation loss or the gradient-targeted transfer modulation improves the detection significantly, underscoring the importance of attaining high-certainty knowledge and synchronizing the learning paces of the two networks. In addition, combining the two designs achieves the best performance, highlighting their complementary nature and collaborative roles in knowledge acquisition and knowledge transfer.

**Inference time comparison.**  Table 5 compares the inference time on the KITTI validation set. Compared with other methods utilizing dense depth maps, our method demonstrates superior efficiency thanks to our designed self-knowledge distillation framework, without utilizing complex network architectures during inference.

## 5 CONCLUSION

In this paper, we point out that previous weakly supervised monocular 3D detection methods either require additional LiDAR point clouds or need paired images from multiple viewpoints or temporal sequences. To overcome these constraints, we propose a weakly supervised monocular 3D object detection approach that is exclusively grounded on single-view image inputs. Central to our approach is a self-knowledge distillation framework, which effectively harnesses the limited depth information within a single-view image with little computational overhead during inference. We further introduce an uncertainty-aware distillation loss and a gradient-targeted transfer modulation strategy, facilitating knowledge acquisition and knowledge transfer. respectively. Finally, extensive experiments demonstrate the effectiveness of our method.

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

# A APPENDIX

## A.1 ADDITIONAL QUALITATIVE RESULTS

We present additional qualitative results on the RGB image and 3D space, as shown in Fig. 5. Our model exhibits accurate predictions in most scenarios, proving its efficacy and robustness.

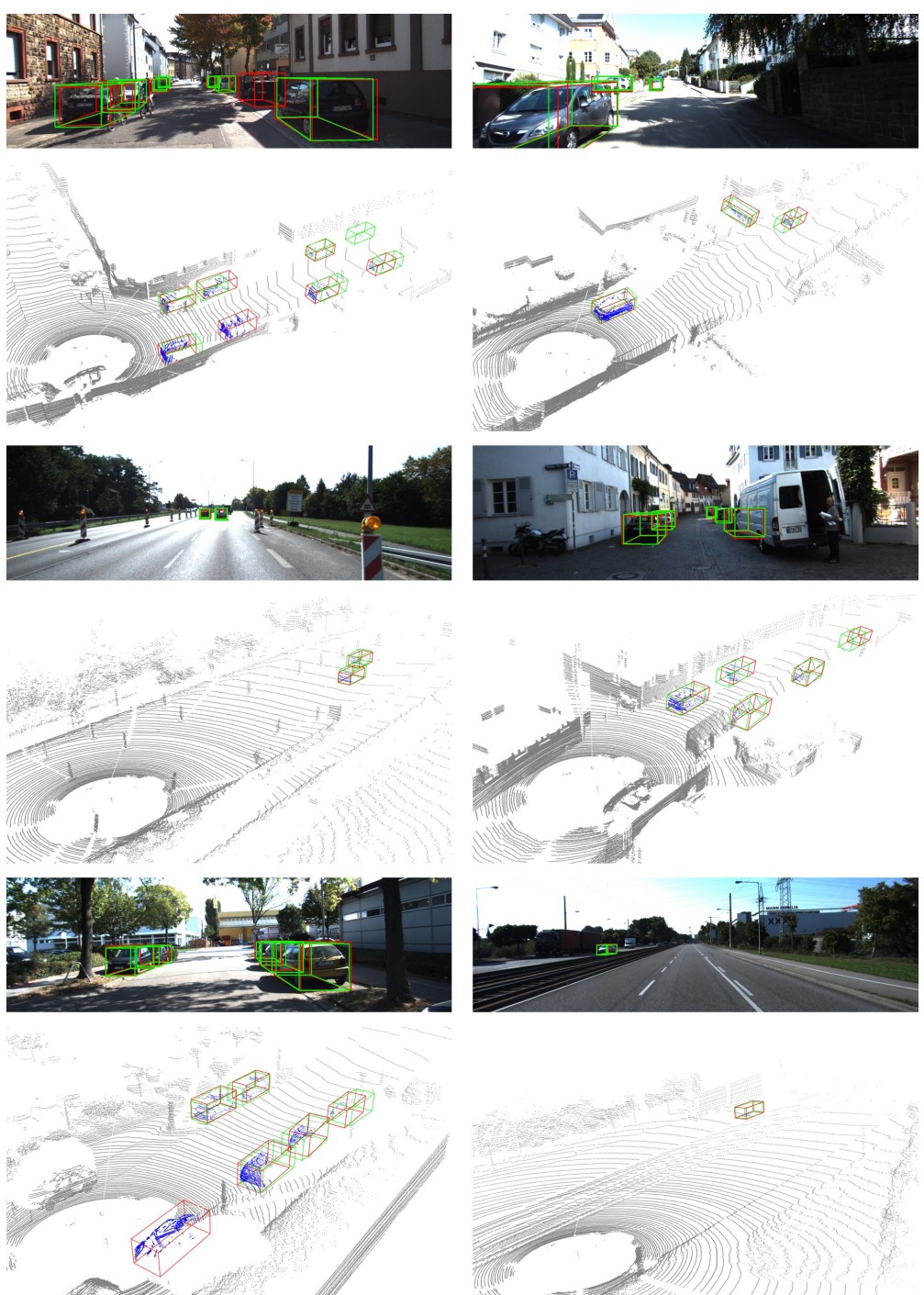

Figure 5: Qualitative illustration on KITTI *val* set. Red boxes denote 3D box ground truth and Green boxes denote our predictions. We can observe that our approach achieves accurate 3D box predictions. Best viewed with zoom-in.

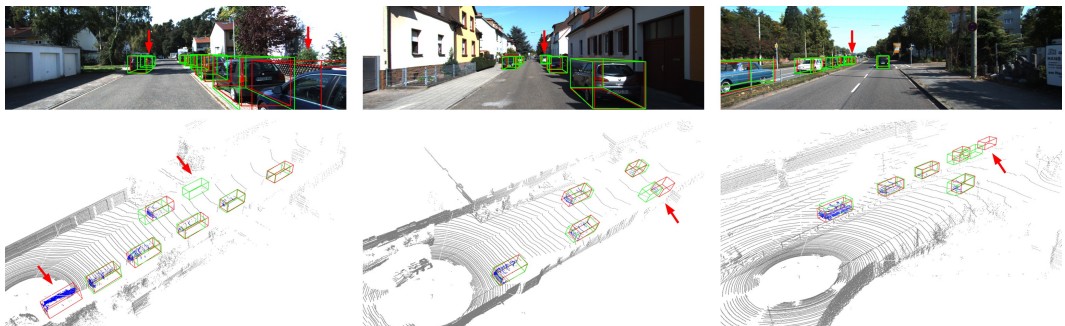

Figure 6: Qualitative illustration of failure cases on KITTI *val* set. Red boxes denote 3D box ground truth and Green boxes denote our predictions. Objects with inaccurate predictions are typically distant, occluded, or partially truncated. Best viewed with zoom-in.

Qualitative results of failure cases are presented in Fig. 6. The proposed SKD-WM3D achieves less accurate performance on challenging objects. It can be observed that SKD-WM3D faces challenges, particularly in cases involving distant (Columns 1, 2, and 3), occluded (Column 1, 3), or truncated objects (Column 1). Further research could be explored under such complex scenarios to enhance performance.

## A.2 DISCUSSIONS

**Difference between our Method and Knowledge Distillation Method.** Our self-knowledge distillation method is different from knowledge distillation methods, which often require a pre-trained teacher network. Our self-knowledge distillation method trains the network progressively to distill and to regularize its own knowledge, without a pre-trained teacher network.

**Compared with Other Knowledge-Distillation-Based M3D Methods.** MonoDistill (Chong et al., 2022) is a fully supervised monocular 3D detection method employing knowledge distillation. In contrast, our novel method, SKD-WM3D, differs from MonoDistill in two key ways. Firstly, SKD-WM3D does not rely on LiDAR point clouds, eliminating the need for costly and complex LiDAR sensors. In contrast, MonoDistill relies on LiDAR point clouds for spatial information. Secondly, the transferred knowledge of MonoDistill has high quality, low uncertainty, and little noise, since the 3D annotations are utilized. However, for the weakly supervised monocular 3D detection task without 3D annotations, directly applying MonoDistill is inappropriate due to the relatively higher uncertainty and more noise in the transferred knowledge. Our method addresses this challenge by introducing an uncertainty-aware distillation loss to enhance robustness in the presence of noisy knowledge.

## A.3 DETAILS OF NETWORK ARCHITECTURE

The self-knowledge distillation framework first takes an RGB image $I$ as input. After feature encoding in the backbone network, we have global features $F_G$. Then the global features $F_G$ are fed into the 2D detection heads and gain 2D head predictions, namely, 2D heatmap $H$, 2D offset $O_{2D}$, and 2D size $S_{2D}$. By combining these 2D head predictions, we can obtain 2D box predictions. The global features $F_G$ are then fed into the depth-guided self-teaching network and the monocular 3D detection network.

**Depth-Guided Self-Teaching Network.** In the depth-guided self-teaching network, the global features $F_G$ are fed into the depth head to produce depth features $F_D$, subsequently employed to generate depth map $D_p$ To ensure the depth-guided self-teaching network acquires depth information, the predicted depth map $D_p$ is supervised by the pseudo ground truth of the depth map $D_{gt}$, adopting the focal loss (Lin et al., 2017) as depth loss $L_{dep}$. The depth features $F_D$ are integrated with the global features $F_G$ via a fusion layer to derive the 3D-like features $F_{G3D}$. Single 3D object features, denoted as $F_{3D}$, are extracted from the 3D-like features $F_{G3D}$ using RoI Align and fed into

Table 6: Ablation study of basic distillation loss. The best results are in **bold**.

| Basic Distillation Loss | $\text{AP}_{BEV}/\text{AP}_{3D}(\text{IoU}= 0.5)|_{R_{40}}$ | | |
| --- | --- | --- | --- |
| | Easy | Moderate | Hard |
| L1 Loss | 53.10/46.04 | 42.64/37.07 | 39.34/34.99 |
| SmoothL1 Loss | **55.47/50.21** | **44.35/41.57** | **41.86/36.92** |

Table 7: Ablation study of different losses.

| Index | $L_{dep}$ | $L_{ud}$ | $\text{AP}_{BEV}/\text{AP}_{3D}(\text{IoU}= 0.5)|_{R_{40}}$ | | |
| --- | --- | --- | --- | --- | --- |
| | | | Easy | Moderate | Hard |
| 1 | ✓ | | 0.68/0.25 | 0.38/0.12 | 0.32/0.09 |
| 2 | | ✓ | 0.00/0.00 | 0.00/0.00 | 0.00/0.00 |
| 3 | ✓ | ✓ | **55.47/50.21** | **44.35/41.57** | **41.86/36.92** |

the depth-aware 3D head. The predicted 3D boxes $\widehat{B}_p^{3D}$ are projected into the 2D image space to obviate the need for 3D box ground truth, following (Tao et al., 2023). We use the L1 loss function for the projection loss, $\widehat{L}_{proj}$. Furthermore, the predicted uncertainty $\widehat{U}$ quantifies the reliability of the predicted 3D boxes $\widehat{B}_p^{3D}$. Specifically, the depth head comprises a sequence of operations: Conv-BN-ReLU-Conv, where Conv denotes 2D convolutional operation, BN denotes batch normalization, and ReLU denotes the rectified linear unit function.

**Monocular 3D Detection Network.** In the monocular 3D detection network, 2D object features, denoted as $F_{2D}$, are extracted from the global features $F_G$ using RoI Align and subsequently input into the 2D-to-3D head. This head transforms the 2D object features $F_{2D}$ to the 3D boxes predictions, represented as $B_p^{3D}$. The supervision of the predicted 3D boxes $B_p^{3D}$ is facilitated by soft labels generated by the depth-guided self-teaching network. This process enables effective knowledge transfer from the depth-guided self-teaching network to enhance 3D localization within the monocular 3D detection network. Finally, the predicted 3D boxes $B_p^{3D}$ are projected into the 2D image space. We employ the L1 loss for the projection loss $L_{proj}$, which is the same as the projection loss utilized in the depth-guided self-teaching network.

## A.4 Details of Loss Functions

In equation 6, $L_{base}$ includes losses for supervising 2D boxes predicted by the 2D head and projected 2D boxes from the 3D box predictions. For losses for supervising 2D boxes predicted by the 2D head, we adopt the loss design from CenterNet (Zhou et al., 2019) for 2D object detection. The 2D heatmap $H$ signifies the approximate object center in the image, while the 2D offset $O_{2D}$ represents the bias from the estimated 2D center. The 2D size $S_{2D}$ corresponds to the height and width of the 2D bounding box. Following CenterNet, we adopt $L_H$, $L_{O_{2D}}$, and $L_{S_{2D}}$ as 2D loss functions $L_{2D\ box}$. The projection losses are computed using L1 loss, denoted as $\widehat{L}_{proj}$ and $L_{proj}$ for the depth-guided self-teaching network and the monocular 3D detection network, respectively.

## A.5 Quantitative Results

**Basic Distillation Loss.** Table 6 compares the basic distillation loss $L_d$ used in the uncertainty-aware distillation loss $L_{ud}$. It can be observed that compared with L1 loss, using SmoothL1 loss yields the best results. This is because, during the early training phase, both networks provide inaccurate predictions, making it impractical to enforce strict consistency between their outputs. The SmoothL1 loss leaves a soft margin when computing the difference between the two 3D boxes.

**Effect of Different Losses.** Table 7 presents the results of ablation studies comparing three different loss functions. In this ablation study, the 2D losses, including $L_H$, $L_{O_{2D}}$ and $L_{S_{2D}}$, and the projection losses, including $\widehat{L}_{proj}$ and $L_{proj}$ are used as default. Solely utilizing the depth loss leads to performance scores of 0.68/0.25, 0.38/0.12, and 0.32/0.09 for the Easy, Moderate, and Hard categories, respectively. These suboptimal results underscore the necessity of the uncertainty-aware

Table 8: Comparison of the performance of the Car category on KITTI *test* set. For all results, we use $AP|_{R_{40}}$ metrics with IoU threshold equals to 0.7. The best results of weakly supervised approaches are in **bold**.

| Method | Supervision | $AP_{BEV}/AP_{3D}(IoU= 0.7)|_{R_{40}}$ | | |
| | | Easy | Moderate | Hard |
|---|---|---|---|---|
| FQNet (Liu et al., 2019a) | Full | 5.40/2.77 | 3.23/1.51 | 2.46/1.01 |
| GS3D (Li et al., 2019) | | 8.41/4.47 | 6.08/2.90 | 4.94/2.47 |
| ROI-10D (Manhardt et al., 2019) | | 9.78/4.32 | 4.91/2.02 | 3.74/1.46 |
| MonoGRNet (Qin et al., 2019) | | 18.19/9.61 | 11.17/5.74 | 8.73/4.25 |
| MonoPSR (Ku et al., 2019) | | 18.33/10.76 | 12.58/7.25 | 9.91/5.85 |
| M3D-RPN (Brazil & Liu, 2019) | | 21.02/14.76 | 13.67/9.71 | 10.23/7.42 |
| MonoDLE (Ma et al., 2021) | | 24.79/17.23 | 18.89/12.26 | 16.00/10.29 |
| Kinematic (Brazil et al., 2020) | | 26.69/19.07 | 17.52/12.72 | 13.10/9.17 |
| MonoEF (Zhou et al., 2021) | | 29.03/21.29 | 19.70/13.87 | 17.26/11.71 |
| AutoShape (Liu et al., 2021) | | 30.66/22.47 | 20.08/14.17 | 15.59/11.36 |
| MonoDistill (Chong et al., 2022) | | 31.87/22.97 | 22.59/16.03 | 19.72/13.60 |
| MonoDETR (Zhang et al., 2023) | | 33.60/25.00 | 22.11/16.47 | 18.60/13.58 |
| WeakM3D (Peng et al., 2022b) | Weak | 11.82/5.03 | 5.66/2.26 | 4.08/1.63 |
| WeakMono3D (Tao et al., 2023) | | 12.31/6.98 | 8.80/4.85 | 7.81/4.45 |
| **SKD-WM3D (Ours)** | | **15.71/8.95** | **10.15/5.54** | **8.08/4.53** |

Table 9: Comparison of the parameters and flops of our proposed method. The tests are carried out on a single NVIDIA V100 GPU.

| Network | Params | FLOPs | Inference Time |
|---|---|---|---|
| Depth-Guided Self-Teaching Network | 24.44M | 161.13G | 80ms |
| Monocular 3D Detection Network | 20.36M | 61.15G | 33ms |

distillation loss, which enables the M3D network to learn knowledge from the depth-guided self-teaching network and to extract intrinsic depth information from images during inference. When uncertainty-aware distillation is employed, zero metrics are achieved, affirming the effectiveness of incorporating depth information for precise 3D object localization. Finally, employing both losses yields the highest performance, emphasizing the significance of depth information and knowledge transfer. In our weakly supervised approach, both the two losses are indispensable, collectively offering comprehensive guidance for model learning.

**Comparison with Fully Supervised Methods.** Table 8 shows our comparison of the KITTI test set against state-of-the-art weakly supervised monocular 3D detection methods and various fully supervised monocular 3D detection approaches. Our approach demonstrates competitive performance compared to certain fully supervised techniques, such as (Liu et al., 2019a; Li et al., 2019; Manhardt et al., 2019), even when 3D annotations are not available.

**Computational Efficiency of Our Proposed Method.** We conduct a comprehensive computational efficiency analysis of our proposed framework. In Table 9, we present network parameters, floating-point operations per second (FLOPs), and inference time for the networks within the self-knowledge distillation framework. Compared with the depth-guided self-teaching network, the monocular 3D detection network requires fewer parameters and smaller FLOPs, obviating the need for additional modules such as a pre-trained depth network or a depth fusion module. Furthermore, since only the monocular 3D detection network is utilized during inference, it exhibits less inference time than the depth-guided self-teaching network, without adding extra computational burdens in inference.

