# OpenReview forum: "Weakly Supervised Monocular 3D Detection with a Single-View Image"
_ICLR.cc/2024/Conference — ICLR 2024 Conference Withdrawn Submission_

### Official Review · Reviewer_r6LJ · 2023-10-29

**Soundness:** 2 fair
**Presentation:** 2 fair
**Contribution:** 2 fair
**Rating:** 5
**Confidence:** 4

**Summary:**

The manuscript introduces a distillation-based training framework named SKD-WM3D for monocular 3D detection using 2D bounding boxes for weak supervision. The framework incorporates two distinct networks: a self-teaching network (DSN) that utilizes a pseudo depth map as input to understand the 3D geometry, and a monocular 3D detection network which learns from the teacher and is employed during inference. The training process is modulated by the addition of an uncertainty-aware distillation loss and a gradient-targeted transfer modulation strategy. Benchmark tests on the 3D KITTI dataset validate the efficacy of the proposed framework and its training methodologies. Codes are not provided or promised.

**Strengths:**

- The suggested framework is based on the knowledge-distillation paradigm and has been modified by the authors to suit the weakly supervised M3D task, using pseudo-depth as input and utilizing readily available depth estimators.
- The introduced uncertainty-based loss reweighting and gradient-targeted transfer modulation strategy show potential effectiveness and could influence subsequent research.
- The method presented sets a SOTA on several settings, indicating its effectiveness.

**Weaknesses:**

- The literature review on self-supervised and weakly supervised paradigms appears to be limited. A clear rationale for the selection of the knowledge distillation method over self-training [1] approaches would be beneficial.
- The experiments section could benefit from a deeper analysis to offer more comprehensive insights to the readers. For example, an explanation for the poorly observed performance when solely using the Monocular 3D Detection Network in the ablation study (Tab. 3) would be valuable.
- It would be informative to include an analysis regarding the training cost after integrating an additional network, DSN.

[1] DQS3D: Densely-matched Quantization-aware Semi-supervised 3D Detection, ICCV 2023

**Questions:**

Several unclear things:

(1) What is the architecture of the Depth-aware 3D head in DSN? Not mentioned.

(2) What is uncertainty mechanism used here? Not mentioned thus not self-contained.

(3) How to match boxes in the distillation loss? Not mentioned.

(4) Why exp.2 in Table.3 work? Using only one network, there is no distillation loss and the 3d detector cannot be trained.

---

### Official Review · Reviewer_mWR5 · 2023-10-30

**Soundness:** 2 fair
**Presentation:** 3 good
**Contribution:** 2 fair
**Rating:** 3
**Confidence:** 5

**Summary:**

This manuscript introduces a new method for weakly supervised monocular 3D detection, by transferring 3D knowledge from pre-trained depth network to detection network. The authors also propose an uncertainty-aware distillation loss and a gradient-targeted transfer modulation strategy to facilitate this transfer process.

**Strengths:**

1. The proposed method is intuitive and sound. Both knowledge transfer and uncertainty design are intuitive and effective components.

2. The writing is good and the figures are clear, which help us to understand the core idea.

3. The ablations are exhaustive and demonstrate the effectiveness of each idea.

4. The results achieve a new state of art on KITTI benchmark.

**Weaknesses:**

1. My main concern is the over-claimed contribution and motivations. The authors claim that the proposed method (SKD-WM3D) does not need LiDAR data/multi-view data. However, the depth network in SKD-WM3D are supervised by depth labels, which come from projected LiDAR points. In other words, the proposed method still requires LiDAR data, though it is another manner. Therefore, I believe that the claims require to be fixed.

2. KITTI is a small dataset, experiments on other large-scale datasets are beneficial.

**Questions:**

Please see the Weaknesses.

---

### Official Review · Reviewer_Be8N · 2023-11-01

**Soundness:** 3 good
**Presentation:** 2 fair
**Contribution:** 3 good
**Rating:** 6
**Confidence:** 4

**Summary:**

The authors address the problem of weakly-supervised monocular 3D object detection (3DOD) where a 3DOD network using single images as input is trained using only 2D bounding box labels. In contrast to previous works relying on additional data during training (LiDAR, multi-view images) the authors propose an approach relying on an off-the-shelf depth network and two networks for 3DOD with different architectures in a self-knowledge distillation framework. Additionally, uncertainty-based loss weighting and synchronized learning via gradient weighting strategies are proposed to enhance the distillation framework. Evaluation on the KITTI dataset shows that the method outperforms previous state-of-the-art approaches.

**Strengths:**

-	The introduction provides a clear and interesting motivation for the considered problem. The differences to previous approaches are described clearly in the introduction and related works sections.
-	Being able to train monocular 3D object detection networks in weakl-supervised fashion without additional data presents an advantage in terms of generalizable applicability over previous methods.
-	The proposed method is able to outperform a previous state-of-the-art methods. The single method components are verified by an ablation study.

**Weaknesses:**

Issues:

1.	It would help the clarity of the method description, if in the beginning a bit more focus could be put on explaining a simple baseline using, e.g., only the monocular 3D detection network and the corresponding loss after the projection. For example, it has not become entirely clear to me, if this baseline would work at all.

2.	It is a bit confusing to talk about a depth-guided network and a monocular 3D detection network if both of these networks are actually doing the same task but just with a different network architecture. That said, I was wondering, if there is a reason except for computational complexity to not use the depth-guided network at inference?

3.	In general it would be nice to gain a bit more insight into the self-distillation scheme. For example, is it important which network is used during inference? Do both networks profit from the scheme? Is there a way to prioritize one of the network or are there observable trade-offs when choosing different hyper parameters? A bit more analysis on these issues would be very interesting.

Minor comments:

4.	It would help to clearly state the type of supervision used, i.e., 2D bounding boxes as far as I understood.

5.	It would be nice to evaluate the method on more than one dataset, e.g, also on the nuScenes dataset, and for more than one network architecture.

6.	It would help to provide a bit more insight about the influence of different hyper parameter configurations on the method through additional ablation studies. For example it is not quite clear how different choices of the soft margin in Equation 4 affect the performance.

7.	Figure 3: Instead of just showing the method output it would be more helpful to show visualizations of a baseline vs. the proposed method. Thereby, one would see which aspects of the predicted bounding boxes are improved.

**Questions:**

-	At the end of Section 2.2, the authors mention that pseudo multi-view perspective suffers from performance degradation. Could the authors explain a bit more detailed why this is not the case for their method using only single images?
-	Do the methods in Table 1 and 2 use the same backbone and image resolution as the authors’ method? Or at least one of comparable complexity? It would help to add this information to the Tables.
-	Can the method be combined with arbitrary architectures for monocular 3D object detection or what might be the limitations? It would be helpful to gain a bit more insight, if the method can be used to enhance arbitrary weakly supervised 3D object detection methods in the future.

---

### Official Review · Reviewer_Wtxb · 2023-11-02

**Soundness:** 3 good
**Presentation:** 2 fair
**Contribution:** 2 fair
**Rating:** 5
**Confidence:** 4

**Summary:**

The paper proposes a self-knowledge distillation framework for monocular 3D detection framework that does not use LiDAR point clouds, multi-view images or 3D annotations. The framework consists of a depth-guided self-training network (DSN) and a monocular 3D detection network (MDN). The DSN uses the depth generated from an off-the-shelf depth estimator to learn 3D aware features. The knowledge of these features can be transferred to MDN. In addition, the paper designs an uncertainty-aware distillation loss for better knowledge transfer and a gradient-targeted transfer modulation strategy to synchronize the learning paces of DSN and MDN. Experiments show that the proposed method outperforms the SOTA weakly supervised monocular 3D detection approach.

**Strengths:**

- The proposed method improves the SOTA in the challenging monocular 3D detection task where no 3D annotations are available.
- The paper provides detailed ablation studies to validate the proposed uncertainty-aware distillation loss and gradient-targeted transfer modulation strategy.

**Weaknesses:**

- Some parts of the paper are missing. (1) The author should provide more information about the depth estimator. What is the design of the depth head and the depth estimator? How to train the off-the-shelf depth estimator? How is the training data generated for the off-the-shelf depth esitimator? (2) How is the uncertainty generated? Is it just the objectness(existence) score of each bounding box?
- Though the paper emphasizes that it does not directly take LiDAR point clouds or multi-view images as input, it seems that it still requires some kinds of 3D data for training the depth estimator (either 2.5 depth/range images or 3D LiDAR point clouds). I think these kind of information might be stronger signals compared to multi-view images. Since the details of the depth estimators are missing, I would like the author to talk more about it. If range images are still required for training the depth estimator, then I think the papers actually shares similar setup with existing baselines, the difference is that the paper does not directly use GT but just pseudo-GT. What is the practical value of using pseudo-GT instead of GT here?
- The paper only evaluates on KITTI. However, now there have been a lot of datasets that are large and of better annotation quality, such as Waymo and nuScenes. It would be better if the author provides experiments on extra dataset.

**Questions:**

See Weaknesses